# Complex Spatio-Temporal Interplay of Distinct Immune and Bone Cell Subsets during Bone Fracture Healing

**DOI:** 10.3390/cells13010040

**Published:** 2023-12-24

**Authors:** Claudia Schlundt, Radost A. Saß, Christian H. Bucher, Sabine Bartosch, Anja E. Hauser, Hans-Dieter Volk, Georg N. Duda, Katharina Schmidt-Bleek

**Affiliations:** 1Julius Wolff Institut, BIH at Charité—Universitätsmedizin Berlin, Augustenburger Platz 1, 13353 Berlin, Germany; c.schlundt@gmx.net (C.S.); radost.sass@charite.de (R.A.S.); christian.bucher@charite.de (C.H.B.); georg.duda@charite.de (G.N.D.); 2BIH Center for Regenerative Therapies, BIH at Charité—Universitätsmedizin Berlin, Augustenburger Platz 1, 13353 Berlin, Germany; hans-dieter.volk@charite.de; 3Berlin School for Regenerative Therapies, Charité—Universitätsmedizin Berlin, Augustenburger Plarz 1, 13353 Berlin, Germany; sabine.bartosch@charite.de; 4Rheumatology and Clinical Immunology, Charité—Universitätsmedizin Berlin, Charitéplatz 1, 10117 Berlin, Germany; hauser@drfz.de; 5Immune Dynamics, Deutsches Rheuma-Forschungszentrum Berlin, Charitéplatz 1, 10117 Berlin, Germany; 6Institute of Medical Immunology, Charité—Universitätsmedizin Berlin, Augustenburger Platz 1, 13353 Berlin, Germany

**Keywords:** bone healing, immune cells, bone cells, hypoxia, revascularization, histology

## Abstract

Background: The healing of a bone injury is a highly complex process involving a multitude of different tissue and cell types, including immune cells, which play a major role in the initiation and progression of bone regeneration. Methods: We histologically analyzed the spatio-temporal occurrence of cells of the innate immune system (macrophages), the adaptive immune system (B and T lymphocytes), and bone cells (osteoblasts and osteoclasts) in the fracture area of a femoral osteotomy over the healing time. This study was performed in a bone osteotomy gap mouse model. We also investigated two key challenges of successful bone regeneration: hypoxia and revascularization. Results: Macrophages were present in and around the fracture gap throughout the entire healing period. The switch from initially pro-inflammatory M1 macrophages to the anti-inflammatory M2 phenotype coincided with the revascularization as well as the appearance of osteoblasts in the fracture area. This indicates that M2 macrophages are necessary for the restoration of vessels and that they also play an orchestrating role in osteoblastogenesis during bone healing. The presence of adaptive immune cells throughout the healing process emphasizes their essential role for regenerative processes that exceeds a mere pathogen defense. B and T cells co-localize consistently with bone cells throughout the healing process, consolidating their crucial role in guiding bone formation. These histological data provide, for the first time, comprehensive information about the complex interrelationships of the cellular network during the entire bone healing process in one standardized set up. With this, an overall picture of the spatio-temporal interplay of cellular key players in a bone healing scenario has been created. Conclusions: A spatio-temporal distribution of immune cells, bone cells, and factors driving bone healing at time points that are decisive for this process—especially during the initial steps of inflammation and revascularization, as well as the soft and hard callus phases—has been visualized. The results show that the bone healing cascade does not consist of five distinct, consecutive phases but is a rather complex interrelated and continuous process of events, especially at the onset of healing.

## 1. Background

Bone fractures continue to be on the rise due to an aging and active society, especially in industrialized countries [1,2,3]. Luckily, bones as organs have the impressive capability to almost completely regenerate after fracture in the majority of otherwise healthy patients. Thus, a better understanding of the cascade of successful bone healing could help to unravel principles of regeneration [4]. This understanding of scar-free bone healing could help to understand the blueprint for tissue regeneration, even in cases where healing would naturally not occur.

So far, we and others have identified different phases of bone fracture healing and divided the complex process into five consecutive but overlapping phases [5,6], rather than the initial four [7]: hematoma formation with an accompanying (1) pro- and (2) anti-inflammatory reaction, (3) soft callus formation, (4) hard callus formation, and (5) the final remodeling step [8].

During this healing cascade, immune cells play a key role [9,10,11]. They are among the first cells infiltrating a fracture zone after injury [12,13]. During the early healing phases, immune cells are responsible, among other aspects, for the clearance of cell debris (e.g., macrophages) and, by secreting signaling molecules, for the recruitment of all other cells necessary for the successful progression of healing [14].

Other research groups and ours have already shown that macrophages are indispensable for the correct guidance of soft and hard callus formation during bone repair and thus successful bone regeneration [15,16,17]. T cells, as part of adaptive immunity, also play a crucial role in bone healing [18,19]. We already showed that an elevated number of CD8+ T cells impair bone repair in mice and humans (in humans: CD8+ terminally differentiated T cells, TEMRA) [20]. Furthermore, using a mouse model system without mature B and T cells (RAG1^−/−^), a change in bone quality was observed: bones of these RAG1^−/−^ mice were stiffer in comparison to the control group and thus were more prone to fracture [21]. Deeper analyses showed a dysregulation of collagen deposition and osteoblast distribution in these mice, which was associated with a lack of T cells rather than B cells. Activated CD4+ T cells have been reported to cause bone loss and joint destruction in a murine arthritis model [22]. Activated T cells express the Receptor Activator of NF-κB Ligand (RANKL), which binds to its receptor RANK on osteoclast precursors and mature osteoclasts and thus induces osteoclast differentiation and activation [23]. A rescue mechanism exists that prevents chronic stimulation of osteoclasts by activated T cells through the simultaneous release of the cytokine interferon γ (IFNγ). IFNγ secretion blocks the activation of osteoclasts induced by the RANKL–RANK pathway. B cells are a main producer of the RANKL decoy receptor osteoprotegerin (OPG). Binding of RANKL to OPG inhibits the activation of RANK on osteoclasts [24]. Thus, B cells are also important for the regulation of bone homeostasis and bone remodeling during bone regeneration [25].

Tightly coupled to anti-inflammation is the process of revascularization, another pre-requirement for successful bone regeneration [26,27]. During injury, blood vessels are cut off, and the whole healing zone lacks the classical supply of nutrients and oxygen; an early fracture hematoma is characterized by a lack of oxygen and nutrients. Surprisingly, immune cells, such as macrophages and T cells, have the ability to survive under hypoxic conditions [28,29]. Macrophages are metabolically less dependent on the local environment and, moreover, can supply essential factors, like certain amino acids and ATP (adenosine tri phosphast), to other cells, like T/B cells. These immune cell functions are essential for the early healing phase in order to initiate all further healing steps, such as cell recruitment and revascularization.

Because of their key role in the healing process, as mentioned above, we analyze here the macrophage subtypes during the healing process in greater detail. As a biomarker of reaction to hypoxia, we studied HIF1a (hypoxia inducible factor 1a) expression. HIF1a is stabilized under hypoxic conditions; therefore, it is important in the early healing phase when the blood supply is interrupted by the injury and oxygen levels are low during hematoma formation [30]. The coagulation process triggers a pro-inflammatory response, and macrophages of the M1 phenotype (CD68+CD80+ cells) appear. M1 macrophages are among the cells that react quickly to the danger signals by switching from aerobic to anaerobic energy supply, a process occurring during this healing phase [16]. The switch from a pro-inflammatory to an anti-inflammatory response is essential for the healing process and can be detected by a phenotype change from M1 to M2 [16]. M2 macrophages were therefore stained by CD68+CD206 co-expression to analyze this process. The next necessary step for successful healing is revascularization of the injured area, and vessel formation was therefore analyzed using the CD105 marker for hematopoietic progenitor cells and endoglin for endothelial cells. B cells (B220) and the distribution of CD4+ and CD8+ T cells were also investigated. B cells are known to be sources for osteoprotegerin and thus influence osteoclastogenesis [24], and the composition of T cells (in particular, the ratio of CD8/CD4 T cells) influences bone healing during all stages, as we have shown previously [10,20,24]. Bone cells were analyzed at all time points; osteoblasts were analyzed via osteocalcin and osteoclasts via cathepsin K.

There are already a multitude of studies that have elucidated the role and occurrence of individual cellular subsets during bone repair. However, to our knowledge, there is still no clear overall picture of the spatio-temporal distribution of different immune and bone cell types directly in the fractured bone, including hypoxic areas and vascularization, during the healing process until the bone is bridged and the remodeling phase begins. In the study presented here, we aimed to unravel the spatio-temporal occurrence of cells of the innate immune system (macrophages of different polarization states), the adaptive immune system (B and T cells), and the bone compartment (osteoblasts and osteoclasts), as well as hypoxia and revascularization during the different phases of bone regeneration, using a non-critical-size mouse osteotomy model.

Our data suggest a strong spatio-temporal interplay between immune cells and bone cells during the bone fracture healing process, and support a complex, intermingling cascade of events rather than a simple, sequential-phase model.

## 2. Materials and Methods

### 2.1. Characterization of the Non-Critical-Size-Defect Mouse Osteotomy Model

Twelve-week-old female C57BL/6N mice (Charles River Laboratories, Nordrhein-Westfalen, Germany), N = 6 per group, were used for this study. Animals were housed in the animal facility of the Charité—Universitätsmedizin Berlin under non-specific, pathogen-free (SPF) conditions (area in the animal facility without additional barrier and filtered air supply) with a controlled temperature (20 ± 2 °C), a light/dark circle of 12 h, and water/food available ad libitum. All mice experiments were performed with ethical permission of the Animal Welfare Act, the National Institutes of Health Guide for Care and Use of Laboratories Animals, as well as in accordance with the National Animal Welfare Guidelines, and this study was approved by the local legal representative of animal rights protection authorities (G0008/12).

### 2.2. Surgical Procedure

The osteotomy was performed in the left femur of the mice, as described before. In brief, mice were anaesthetized through isoflurane inhalation and received a subcutaneous injection of Buprenorphine (0.03 mg/kg) and Clindamycin (45 mg/kg). After shaving the operation field of the left femur, the skin was opened, and the femur was carefully exposed through a blunt preparation of the surrounding muscles. An external fixator (RISystem, Davos, Switzerland) was mounted on the femur, and the osteotomy was set with a gap size of 0.7 mm. This fixator device offers a very stable fixation, and we used the 100% bar. Afterwards, the wound was sutured, and mice received pain treatment by supplementing the drinking water with tramadol hydrochloride (25 mg/L) for three days.

### 2.3. Sample Collection

Before sample collection, mice were euthanized through an intraperitoneal injection of a mixture of Ketamine (60 mg/kg) and Medetomidine (0.3 mg/kg), followed by cervical dislocation. Osteotomized femora were collected for histological and immunohistological analysis.

### 2.4. Histological and Immunohistological Analysis

All osteotomized femora were cryo-embedded for histological and immunohistological analysis based on the method of Tadafumi Kawamoto [31]. The advantage of this technique is that the bones do not have to be decalcified before embedding. Harvested bones were directly fixed in 4% paraformaldehyde for 4 h at 4 °C. Afterwards, they went through increasing sucrose solutions of 10%, 20%, and 30%, respectively, with 24 h in each at 4 °C. The embedding procedure was performed in SCEM media (SectionLab, Yokohama, Japan) in pre-cooled n-hexane. Embedded bones were kept at −80 °C until they were cut. The cutting was performed with a cryostat (Leica), and 7 µm thin serial sections were produced, which were further stained using Movat’s Pentachrome or immunohistology, respectively (Figure 1).

Movat’s Pentachrome staining was performed as follows. Frozen cryo-sections were brought to room temperature, followed by rehydration for 20 min in 1× phosphate buffered saline (PBS). Sections were placed for 3 min in 3% acetic acid, followed by a 30 min staining in an alcian blue solution in 3% acetic acid. Sections were rinsed in 3% acetic acid, followed by a rinsing step in distilled water. Afterwards, the staining was continued as follows. Sections were placed for 1 h in ethyl alcohol, for 10 min in tap water, for 10 min in iron hematoxylin, for 10 min in tap water, and for 15 min in brilliant crocein acid fuchsine, and then rinsed in 0.5% acetic acid. An incubation step for 20 min in 5% phosphotungstic acid followed. Subsequently, sections were placed in 0.5% acetic acid for 1 min, 3× for 2 min in 96% alcohol, for 1 h in saffron du gâtinais, and again 3× for 2 min in 96% alcohol. Finally, sections were placed 2× for 5 min in xylol and embedded with vitro clud.

For the immunohistological analysis, frozen sections were thawed and rehydrated for 20 min in 1× PBS. The following steps were performed in a humified chamber at room temperature in the dark.

Staining of CD4+ T cells, CD8+ T cells, and osteoblasts: Rehydrated sections were blocked for 1 h in 1× tris-buffered saline (TBS) + 10% fetal bovine serum (FBS) + 0.2% Tween-20. Tissue sections were stained for osteocalcin (Enzo Life Sciences, Lörrach, Germany) for 1 h, followed by a washing step in TBS-T (TBS + 5% FBS + 0.2% Tween-20). A secondary anti-rabbit antibody (Invitrogen, Henningsdorf, Germany) was applied for 1 h to the section to visualize the signal of osteocalcin. After a washing step, the staining for CD4 (in-house production of Deutsches Rheuma-Forschungszentrum) followed for 1 h. Subsequently, sections were washed and stained for anti-rat (Invitrogen) as the secondary antibody for CD4. In the next step, tissue samples were washed and stained for CD8 (Abcam, Berlin, Germany) for 1 h, washed, stained for DAPI (Invitrogen, 10 min in 1× PBS), and embedded in fluorescence mounting medium.

Staining for B cells and osteoclasts: After thawing and rehydration, tissue samples were blocked in 1× PBS + 10% FBS + 10% rat serum and 0.1% Tween-20. Tissue sections were stained simultaneously for B220 (B cell marker; in-house production of Deutsches Rheuma-Forschungszentrum) and Cathepsin K (osteoclast marker; Abcam) for 1 h diluted in 1× PBS + 10% FBS and 0.1% Tween-20 (PBS-T). Sections were washed and stained for 1 h for anti-rabbit to visualize the staining of Cathepsin K. After washing, tissue samples were stained for DAPI (10 min in 1× PBS), washed again, and embedded in fluorescence mounting medium.

Staining of macrophage subsets: Rehydrated sections were blocked for 1 h in TBS + 5% FBS + 10% rat serum and 0.1% Tween-20. Afterwards, sections were simultaneously stained for CD68 (AbD Serotec, Neuried, Germany), CD80 (Biolegend, Amsterdam, The Netherlands), and CD206 (Biolegend) for 1 h in TBS + 5% FBS + 0.1% Tween-20 (TBS-T). Sections were washed in TBS-T and stained for 10 min with DAPI to visualize the cell nuclei. Subsequently, sections were washed in 1× PBS and embedded in fluorescence mounting medium.

Staining of hypoxic areas and newly formed vessels: Hypoxic areas were stained with HIF1α (Novus Biologicals, Wiesbaden, Germany) and endothelial cells were stained with CD105 (eBioscience, Frankfurt am Main, Germany). Sections were thawed, rehydrated, and blocked in TBS + 0.2% Tween-20 for 1 h. Then, the endothelial cells were stained using CD105 antibodies for 1 h, followed by a washing step and the staining of the corresponding secondary antibody anti-rat for 1 h. Tissue sections were washed and stained for HIF1α for 1 h. After washing, an anti-rabbit as the secondary antibody for the visualization of HIF1α was added to the sections. After washing, DAPI staining followed. Sections were washed and embedded in fluorescence mounting media. Stained sections were analyzed with a confocal laser scanning microscope 710 (LSM710, Carl Zeiss, Oberkochen, Germany).

### 2.5. Histomorphometry

Histomorphometric analysis was performed with Movat’s Pentachrome stainings. Therefore, a region of interest (ROI) was defined that included the osteotomy gap (0.7 mm) and 0.35 mm on either side of the osteotomy gap. Thus, the ROI encompassed 1.4 mm with a center in the middle of the osteotomy gap. A MACRO was programmed specifically for this study to analyze and evaluate the total callus area, cartilage, and newly mineralized bone in the histological images. These parameters were calculated as a proportion of the area of the total callus within the predefined ROI. This was necessary because the width differs in different healing stages.

### 2.6. Statistics

The statistical evaluation of the presented data was performed with the statistics program SPSS (IBM). Data are presented as boxplots. A normal distribution of the data was excluded due to the small sample size. Therefore, the Mann–Whitney U test was applied for the statistical evaluations. The significance value *p* was set at *p* ≤ 0.05 for statistical significance. The Bonferroni correction was used, and an adaptation of *p* ≤ 0.05/n (n = number of compared samples) was performed.

## 3. Results

### 3.1. Characterization of the Different Healing Phases in a Non-Critical-Size-Defect Mouse Osteotomy Model

A non-critical-size-defect mouse osteotomy model was chosen for the in vivo characterization of the distribution of different skeletal and immune cell types, as well as cellular responses to hypoxia and revascularization. In the first part of this study, healing of the femora was monitored over a period of 21 days through histology using Movat’s Pentachrome stainings (Figure 2). A hematoma formed in the first 2 days after the osteotomy, which was replaced by fibrous tissue in the fracture gap after around 3 days (Figure 2, day to 1 to 3). The formation of cartilaginous tissue started between day 7 and 14 and resulted in a cartilage anlage filling the osteotomy gap (soft callus formation) (Figure 2, day 7 to 14). The soft callus was gradually replaced by newly formed woven bone, which eventually underwent a remodeling process and restored the original shape of the femur (Figure 2, day 21). Bone regeneration in the mouse osteotomy model chosen here was therefore shown to follow the classical steps of endochondral fracture healing.

By means of histomorphometrical analysis of the Movat’s Pentachrome images’ total callus area, the percentage of cartilage and percentage of mineralized bone were investigated. The total callus area increased in the first 3 days, followed by a continuous decrease until day 21 (Figure 3A). The cartilage peaked 14 days post-osteotomy, whereas almost no cartilage was visible in the fracture zone at the other time points (Figure 3B). A marked increase in mineralized bone tissue was observed from day 7 through day 14 to day 21 (Figure 3C).

### 3.2. Spatio-Temporal Distribution of Distinct Immune and Bone Cell Subsets in the Osteotomy Area

Spatio-temporal studies of representative molecules and cells involved in bone regeneration were carried out by means of immunofluorescence analyses in order to monitor the essential phases of bone healing [26], especially the initial healing processes (the staining strategy is displayed in Figure 1). Quantifying the specific analyzed immune cells within the callus area revealed rather high standard deviations and a great fluctuation over the time course of healing (Figure 4).

Macrophages exhibited a pro-inflammatory M1 phenotype during the initial bone repair phase (Figure 5), which switched to the anti-inflammatory M2 phenotype approximately 3 days post-osteotomy (Figure 7). Then, 21 days after the osteotomy, when the hard callus had already formed and even the remodeling phase began, we detected predominantly CD68 single positive macrophages in the former osteotomy area (Figure 10). CD68 was used as a pan-macrophage marker, M1 was defined as CD68+CD80+, M2 was defined as CD68+CD206+, and CD68+CD80-CD206- was defined as non-activated MΦ macrophages (Appendix A: Callus images for each time point: MΦ, M1, M2).

One day after the osteotomy, HIF1α+ cells could be detected within the fracture zone. At the same time, the first CD105+ endothelial cells became visible at the outer edges of the osteotomy gap (Figure 5, Hif1α, CD105). Although HIF1α+ cells disappeared 3 days post-osteotomy, they reappeared near the former osteotomy zone 7 days post-surgery (Figure 8, HIF1α). At the later time points, no more HIF1α+ cells could be detected in the investigated area. The first vessel-like structures, identified by CD105+ cells, were detectable at the periosteum 3 days post-osteotomy and also at the endosteum 7 days after osteotomy (Figure 7 and Figure 8, CD105). At 14 days after surgery, vessels were located around the cartilage, which filled the former osteotomy gap (Figure 9, CD105). At day 21, vessels became visible at the periosteum and in the newly formed bone marrow cavity (Figure 10, CD105) (Appendix A: Callus images for each time point: Hif1a and CD105).

In the early fracture healing phase, osteoblasts were detected at the periosteum, but also in the bone marrow around the osteotomy area (Figure 5, Figure 6 and Figure 7, OC). The osteoblasts partially showed direct co-localization with CD4+ and CD8+ T cells in the bone marrow (Figure 5, Figure 6 and Figure 7, OC, CD4, CD8). Osteoblasts were also detected directly in the osteotomy area and on the endosteal side of the cortices 7 days post-osteotomy, partially co-localized with CD4+ and CD8+ T cells (Figure 8, OC, CD4, CD8). Then, 14 and 21 days post-surgery, osteoblasts were predominantly visible at the endosteum and periosteum (Figure 9 and Figure 10; Appendix A: Callus images for each time point: CD4, CD8, OC). Osteoclasts and B cells were also found in or around the fracture gap over the entire 21-day period. In the first 2 days, osteoclasts were predominantly visible at the periosteum, but also in the bone marrow around the osteotomy gap (Figure 5 and Figure 6, CK, B220). Here, they partially co-localized directly with B220+ B cells. At days 3 and 7 post-surgery, osteoclasts were also found at the endosteal side of the cortical bone and directly in the osteotomy area, respectively (Figure 7 and Figure 8, CK, B220). B cells were detected in the bone marrow adjacent to the fracture gap throughout the entire investigated healing period. In line with the findings of the first 2 days, we could also partially observe a direct co-localization of B cells with osteoclasts at each time point analyzed (Appendix A: Callus images for each time point: B220, CK). A schematic summary of the cellular distribution of the analyzed cell types and tissues is shown in Figure 11.

**Figure 5 cells-13-00040-f005:**
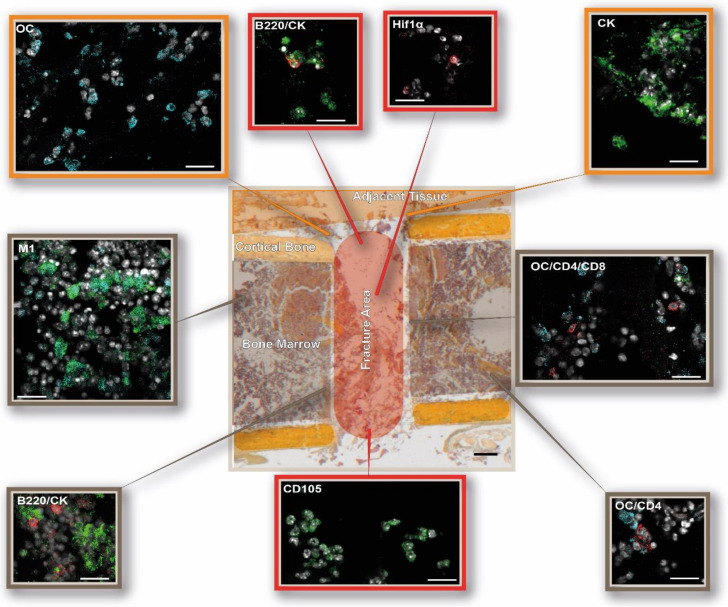
DAY 1—The osteotomy hematoma is characterized by cellular responses to hypoxia and M1 macrophages 1 day post-surgery. Schematic overview of the immune and bone cell distribution in and around the osteotomy gap. The following cellular subsets were investigated: osteoblasts (osteocalcin, OC: light blue), CD4+ T cells (CD4: red), and CD8+ T cells (CD8: green); osteoclasts (cathepsin K, CK: green) and B cells (B220: red); macrophages (MΦ, CD68: green; M1, CD86: light blue; M2, CD206: red), hypoxia (HIF1α: red), and endothelial cells (CD105: green). Shown are representative immunofluorescence images according to their location in the examined bone area. The schematic overview illustration was generated using the Movat’s Pentachrome staining for day 1 from Figure 2. Scale bar: 200 µm overview, 20 µm (outtake = sections of the different stainings/cell subsets). Representative images from N = 4 osteotomized mice.

**Figure 6 cells-13-00040-f006:**
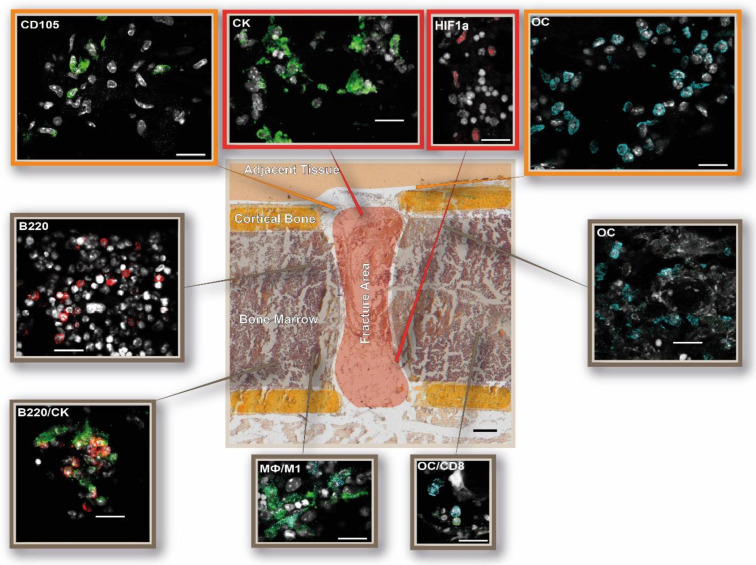
DAY 2—The osteotomy hematoma is characterized by cellular responses to hypoxia and M1 macrophages 2 days post-surgery. Schematic overview of the immune and bone cell distribution in and around the osteotomy gap. The following cellular subsets were investigated: osteoblasts (osteocalcin, OC: light blue), CD4+ T cells (CD4: red), and CD8+ T cells (CD8: green); osteoclasts (cathepsin K, CK: green) and B cells (B220: red); macrophages (MΦ, CD68: green; M1, CD86: light blue; M2, CD206: red), hypoxia (HIF1α: red), and endothelial cells (CD105: green). Shown are representative immunofluorescence images according to their location in the examined bone area. The schematic overview illustration was generated using the Movat’s Pentachrome staining for day 2 from Figure 2. Scale bar: 200 µm overview, 20 µm outtake. Representative images from N = 4 osteotomized mice.

**Figure 7 cells-13-00040-f007:**
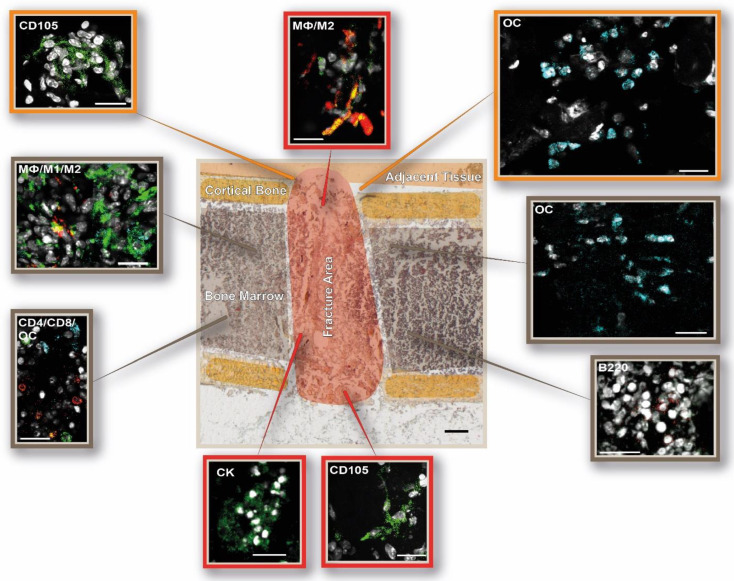
DAY 3—Anti-inflammatory M2 macrophages start to infiltrate the osteotomy area, and the first vessel-like structures are formed 3 days post-surgery. Schematic depiction of the immune and bone cell distribution in and around the osteotomy gap. The following cellular subsets were investigated: osteoblasts (osteocalcin, OC: light blue), CD4+ T cells (CD4: red), and CD8+ T cells (CD8: green); osteoclasts (cathepsin K, CK: green) and B cells (B220: red); macrophages (MΦ, CD68: green; M1, CD86: light blue; M2, CD206: red), hypoxia (HIF1α: red), and endothelial cells (CD105: green). Shown are representative immunofluorescence images according to their location in the examined bone area. The schematic overview illustration was generated using the Movat’s Pentachrome staining for day 3 from Figure 2. Scale bar: 200 µ overview, 20 µm outtake. Representative images from N = 4 osteotomized mice.

**Figure 8 cells-13-00040-f008:**
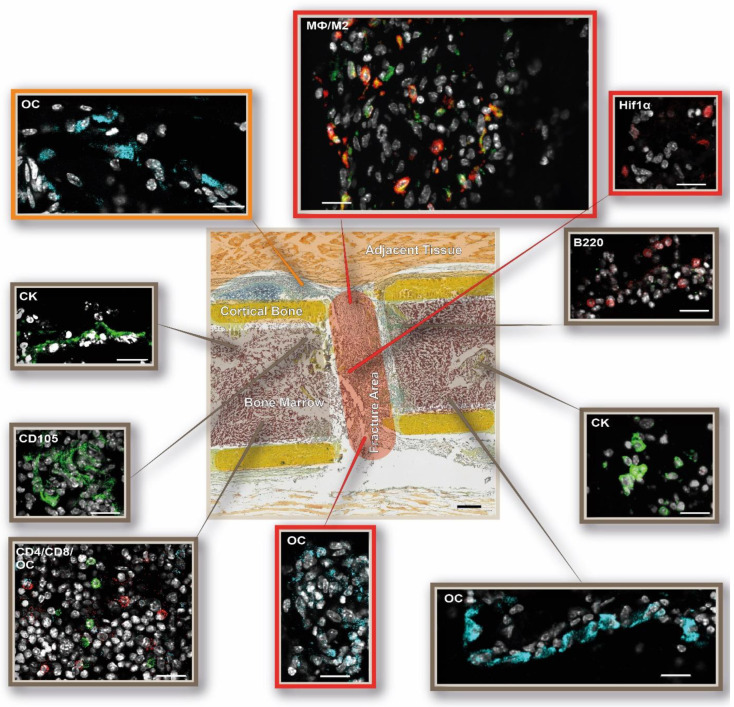
DAY 7—The osteotomy gap is characterized by M2 macrophages and infiltrating osteoblasts 7 days post-surgery. Schematic depiction of the immune and bone cell distribution in and around the osteotomy gap. The following cellular subsets were investigated: osteoblasts (osteocalcin, OC: light blue), CD4+ T cells (CD4: red), and CD8+ T cells (CD8: green); osteoclasts (cathepsin K, CK: green) and B cells (B220: red); macrophages (MΦ, CD68: green; M1, CD86: light blue; M2, CD206: red) and endothelial cells (CD105: green). Shown are representative immunofluorescence images according to their location in the examined bone area. The schematic overview illustration was generated using the Movat’s Pentachrome staining for day 7 from Figure 2. Scale bar: 200 µm overview, 20 µm outtake. Representative images from N = 4 osteotomized mice.

**Figure 9 cells-13-00040-f009:**
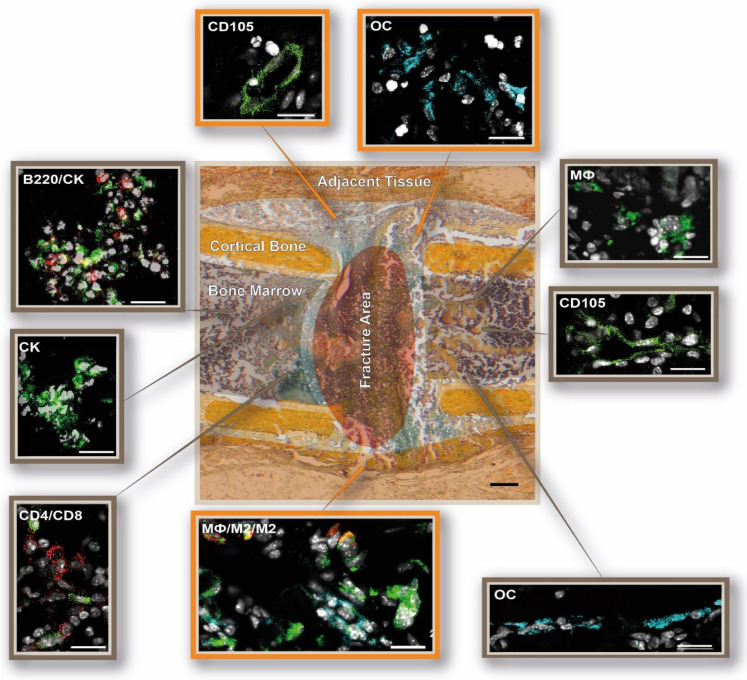
DAY 14—The fracture gap is predominantly filled with cartilage tissue 14 days post-surgery. The following cellular subsets were investigated: osteoblasts (osteocalcin, OC: light blue), CD4+ T cells (CD4: red), and CD8+ T cells (CD8: green); osteoclasts (cathepsin K, CK: green) and B cells (B220: red); macrophages (MΦ, CD68: green; M1, CD86: light blue; M2, CD206: red) and endothelial cells (CD105: green). Shown are representative immunofluorescence images according to their location in the examined bone area. The schematic overview illustration was generated using the Movat’s Pentachrome staining for day 14 from Figure 2. Scale bar: 200 µm overview, 20 µm outtake. Representative images from N = 4 osteotomized mice.

**Figure 10 cells-13-00040-f010:**
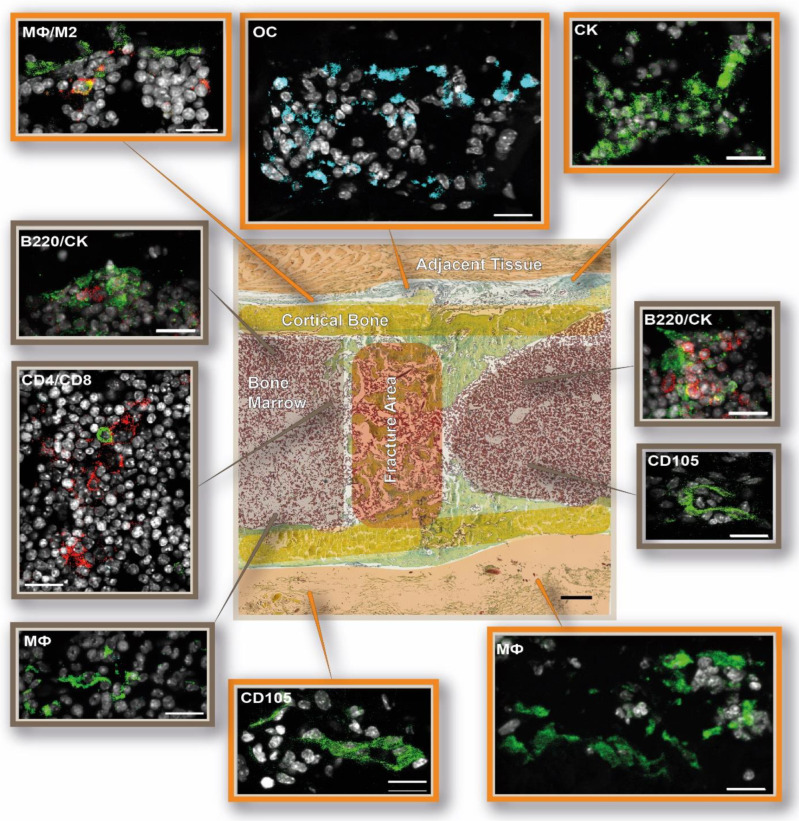
DAY 21—Rebuilding of the former medullary cavity 21 days post-surgery. Visualization of the immune and bone cell distribution locally in and around the osteotomy gap. The following cellular subsets were investigated: osteoblasts (osteocalcin, OC: light blue), CD4+ T cells (CD4: red), and CD8+ T cells (CD8: green); osteoclasts (cathepsin K, CK: green) and B cells (B220: red); macrophages (MΦ, CD68: green, M1; CD86: light blue; M2, CD206: red) and endothelial cells (CD105: green). Shown are representative immunofluorescence images according to their location in the investigated bone area. The schematic overview illustration was generated using the Movat’s Pentachrome staining for day 21 from Figure 2. Scale bar: 200 µm overview, 20 µm outtake. Representative images from N = 4 osteotomized mice.

**Figure 11 cells-13-00040-f011:**
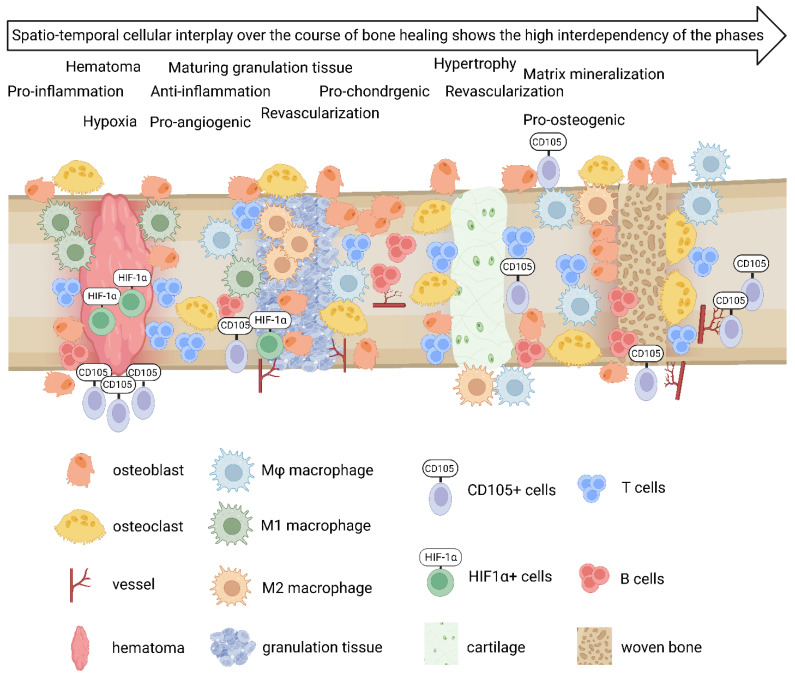
Schematic summary of the distribution of immune and bone cells, revascularization, and hypoxia during the different healing phases of bone regeneration based on the evaluation of the immunofluorescence analysis.

## 4. Discussion

Bone fracture healing is a highly complex process involving different tissues and cell types [7]. For successful bone regeneration, the distinct but overlapping healing phases have to be well-organized and regulated [32]. The data from this study provide a comprehensive, descriptive analysis of the spatio-temporal localization of different immune and bone cell subsets (immune cells: macrophages of different polarization states, CD4+ and CD8+ T cells, B cells; bone cells: osteoblasts and osteoclasts) throughout the healing of a femur fracture with a stable fixation in a mouse osteotomy model [33]. We also included Hif-1a as a marker for the cellular response to hypoxia and endothelial cell markers as an indication of revascularization in our study, as they are key elements of successful regeneration [34,35].

The chosen mouse model was an established osteotomy model system, which recapitulates the classical healing phases of secondary (indirect) bone regeneration with their characteristic tissue formation [36]. Here, the formation of the hematoma occurs in the early fracture healing phase (1–3 days post-surgery), which is subsequently replaced by granulation tissue (3–7 days post-surgery), fibro-cartilage tissue (7–14 days post-surgery), cartilage tissue (14 days post-surgery), and, finally, newly formed woven bone (21 day post-surgery) [37,38].

Hematoma formation is an indispensable step in bone fracture healing [39,40,41,42]. The Movat’s Pentachrome images of the first 3 days after the osteotomy showed the formation of a hematoma in this mouse model. The vessel disruption caused by the osteotomy led to a drop in oxygen and thus to hypoxia [43,44]. HIF1α-positive cells were detectable early on (1 day post-surgery), indicating oxygen deprivation after vascular disruption. HIF1α initiates the expression of pro-angiogenic factors, such as Vascular Endothelial Growth Factor (VEGF) [45], and thus actively supports revascularization as a response to the oxygen depletion [46]. Interestingly, pro-angiogenic processes were already observed one day after surgery, as indicated by the detection of CD105+ cells around the osteotomy gap and in the surrounding tissue. At the same time, predominantly pro-inflammatory M1 macrophages were found around the osteotomy gap, indicating the early pro-inflammatory phase of bone regeneration required to initiate the healing cascade [47]. For successful bone regeneration, the first pro-inflammatory phase must be stopped in a timely and well-regulated manner and replaced by an anti-inflammatory phase. We and others have previously shown that the initiation of anti-inflammation is an indispensable step for the revascularization of the fracture area [48,49]. Accordingly, 3–7 days after the osteotomy, we observed the occurrence of anti-inflammatory M2 macrophages that appeared to infiltrate from the surrounding tissue, coinciding with a loss of the HIF1α signal and the appearance of vessel-like structures 3 days after the osteotomy. The disappearance of the HIF1α signal 3 days post-surgery indicates that the oxygen supply in the fracture area has already been restored. Seven days post-osteotomy, the fracture gap was predominantly colonized by M2 macrophages, but also by osteoblasts. It has already been reported that anti-inflammatory macrophages secrete pro-osteogenic factors and therefore promote osteogenesis and bone formation [50], which supports our findings. M1 and M2 macrophages also play an important role in neovascularization [51]. Thus, the literature is consistent with our observation that M2 macrophages appear in and around the fracture gap simultaneously with the first appearance of vessel-like structures. Interestingly, HIF1α+ cells reappeared 7 days after injury. In addition, the macrophage plasticity steering the healing process is significantly affected by the inflammation of the adaptive immune system, where classically activated M1 macrophages persist longer as the predominant phenotype in the early healing phase [11]. M1 macrophages secrete high amounts of VEGF, thus inducing the re-vascularization of the fracture gap; however, M2 macrophages stabilize the newly formed vessels by secreting growth factors like basic fibroblastic growth factor (bFGF) or platelet-derived growth factor-BB (PDGF-BB) [13,51]. The tight switch in macrophage phenotypes is key for a successful healing process, as shown by the re-vascularization in the early healing phase [52]. Between 7 and 14 days post-osteotomy, the osteotomy area is filled with fibro-cartilaginous tissue, indicating that the soft callus formation is taking place. Cartilage is an avascular tissue consisting only of chondrocytes [53]. The reappearance of hypoxic signals in this phase indicates the beginning of cartilage formation [54,55]. Cartilage serves as a template for the subsequent formation of new woven bone during skeletal regeneration [56]. The Movat’s Pentachrome image as well as the immunohistological analysis confirmed the formation of a cartilage anlage 14 days post-osteotomy. Furthermore, only chondrocytes could be observed within the cartilaginous soft callus, while vessels, B cells, T cells, macrophages, and osteoblasts were detected around the osteotomy gap [57]. During early fracture healing, osteoblasts were predominantly present at the periosteum, a tissue considered to be a source of osteogenic precursor cells [58]. Osteoblasts were also visible in the endosteal area during the soft callus phase. This observation could indicate that new bone formation to repair the osteotomy gap is already taking place. Osteoclasts were detected on both sides of the cortices right at the beginning of healing [59]. This is consistent with the theory that osteoclasts are involved early in the healing process by removing disintegrated bone from the fracture ends to ensure that only healthy bone remains in the healing zone. It is already known that B and T cells are necessary to correctly guide bone formation during skeletal regeneration [21,24], which was confirmed by our data: (1) from the beginning of the healing process, T and B cells were detectable In the bone marrow around the osteotomy gap, and (2) both cell types were partially co-localized with osteoblasts (T cells) and osteoclasts (B cells), respectively. After 21 days, the bone marrow cavity was already partially rebuilt, which was characterized by an equal distribution of T and B cells as well as vessels. In addition, predominantly “non-polarized” MΦ macrophages were detectable. If woven bone was still present in the former osteotomy gap, osteoblasts and osteoclasts were found around it, remodeling the woven bone to restore the former structure of the medullary cavity and thus of the osteotomized bone.

Our histological and immunohistological evaluation provides a hitherto unique overall picture of the spatial and temporal distribution of important immune and bone cell types in the osteotomy area of a regenerating bone over all healing phases. Our study also shows when and where hypoxic cells appear in the fracture area and disappear with the onset of revascularization. This overall picture clearly demonstrates the high interconnectivity between cells of the bone and the immune system during bone regeneration. It further shows and confirms the necessary switch from an initially pro-inflammatory to an anti-inflammatory immune milieu as well as the indispensable revascularization of the osteotomy gap at the beginning of healing, but also in the later stage during hard callus formation.

## 5. Conclusions

Various studies have already reported on individual processes within the healing bone, from which the overall picture of bone healing known today emerges. However, the present analysis provides a spatio-temporal resolution of cells and factors formerly identified as being important for the bone healing process, thus enabling an improved understanding of the cellular processes within the fracture callus throughout the consecutive healing phases. The hematoma formation after injury correlates with the subsequent occurrence of hypoxia (HIF1a) and the pro-inflammatory response (M1). Activation of the revascularization (CD105) and downregulation of the pro-inflammatory response (M2) occur shortly thereafter. Interestingly, M2 macrophages seem to invade the fracture region from the surrounding soft tissue.

The reappearance of HIF1a and the change towards cartilage tissue indicate low oxygen levels involved in cartilage induction. The soft callus itself lacks all cells, but cartilage processes in the surrounding bone marrow are very active. This analysis also verifies that the periosteum is a key tissue in bone formation, as osteoblasts were detected here at a very early stage of the healing process. The role of osteoclasts as phagocytes in early healing was confirmed, and the interdependence of the healing cascade and adaptive immunity once again became very clear. While phases are still determinable in the bone healing process, this study also shows that the process is not a step-by-step but rather a continuous event, and it is a highly interactive, very well-orchestrated cascade.

## Figures and Tables

**Figure 1 cells-13-00040-f001:**
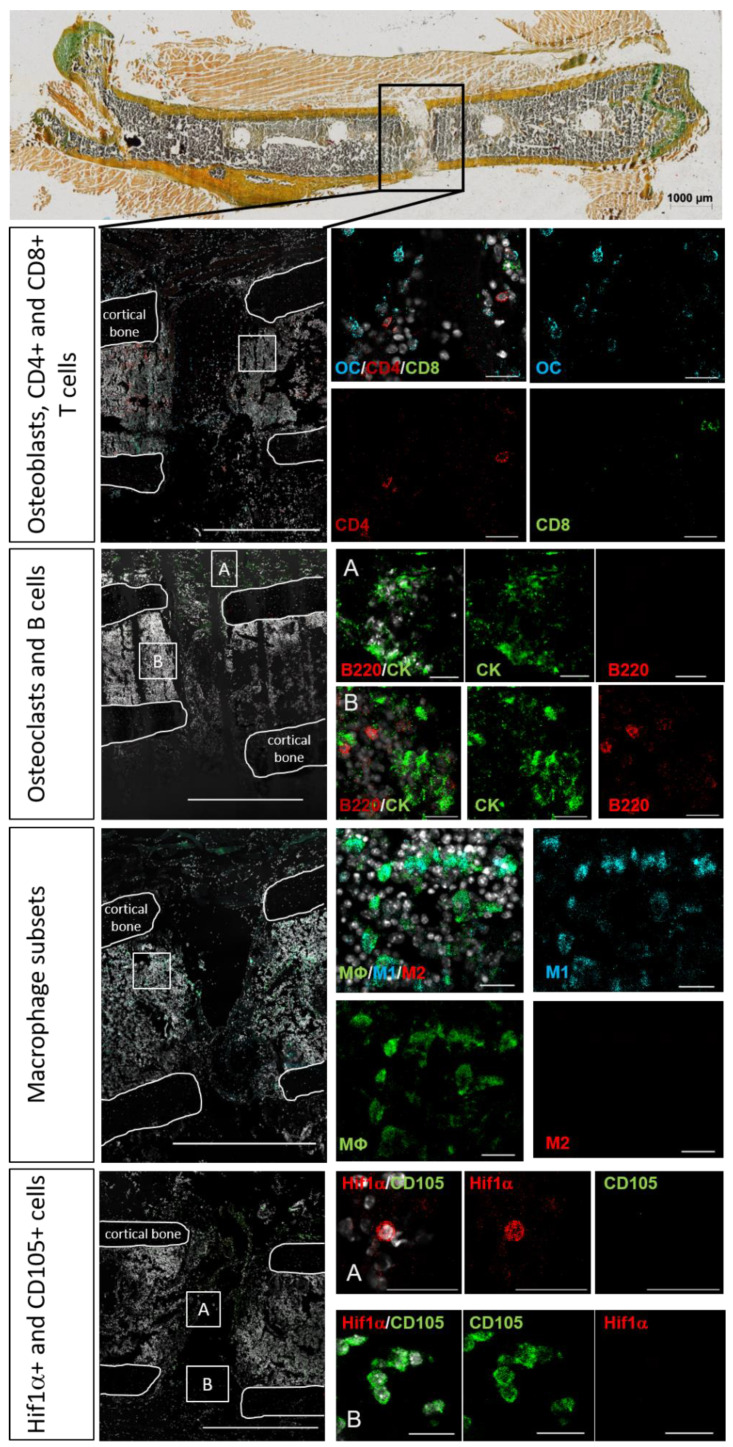
Strategy for the evaluation of the local immune and bone cell composition in and around the osteotomy gap, explained with 1-day-post-surgery images. Four different combinations of immunohistochemical stainings were performed on consecutive bone sections, as depicted on the left side. Displayed is one representative staining of each combination. The white lines indicate the 4 ends of the cortices (cortical bone) as well as the area(s) that is/are enlarged on the right. On top is one representative Movat Pentachrome picture 2 days post-osteotomy, showing the area in and around the osteotomy gap outlined in black, which was analyzed through immunohistochemistry. The following cellular subsets were investigated: osteoblasts (osteocalcin, OC: light blue), CD4+ T cells (CD4: red), and CD8+ T cells (CD8: green); osteoclasts (cathepsin K, CK: green) and B cells (B220: red); macrophages (MΦ, CD68: green; M1, CD86: light blue; M2, CD206: red), hypoxia (HIF1α: red), and newly formed vessels (CD105: green). Colors of the Movat’s Pentachrome staining: yellow = mineralized bone and cartilage; blue/green = cartilage; bluish = connective tissue; purple = bone marrow; black = cell nuclei. Scale bar: 200 μm (enlarged immunohistochemical pictures of the fracture gap) and 20 µm (sections of the different stainings/cell subsets), respectively.

**Figure 2 cells-13-00040-f002:**
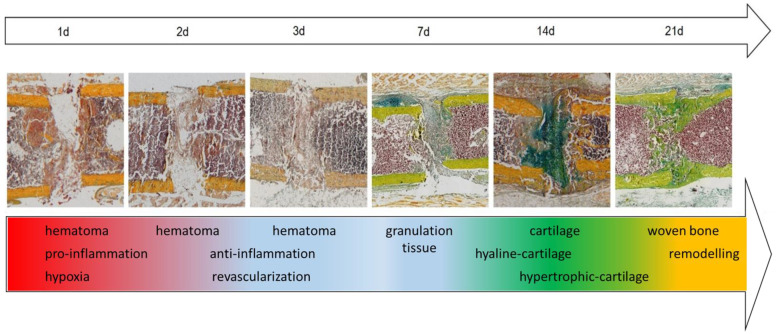
Representative histological images of the different healing phases using Movat’s Pentachrome staining of the osteotomized femora over a time course of 21 d (d = days). The chosen mouse osteotomy model recapitulates the classical phases of endochondral bone fracture healing. Colors of the Movat’s Pentachrome staining: yellow = mineralized bone and cartilage; blue/green = cartilage; bluish = connective tissue; purple = bone marrow; black = cell nuclei.

**Figure 3 cells-13-00040-f003:**
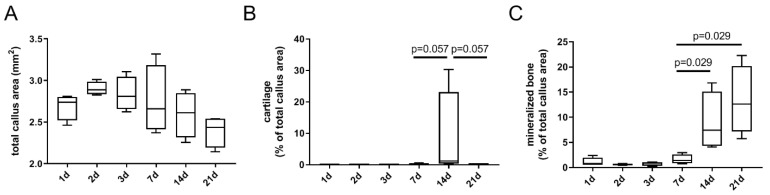
Histomorphometrical evaluation of the Movat’s Pentachrome staining confirms the observed characteristic tissue distribution across the healing cascade. The following parameters were investigated: total callus area (**A**), percentage of cartilage (**B**), and percentage of mineralized bone (**C**). Investigated time points were 1 day (d), 2 d, 3 d, 7 d, 14 d, and 21 d post-osteotomy. n = 4.

**Figure 4 cells-13-00040-f004:**
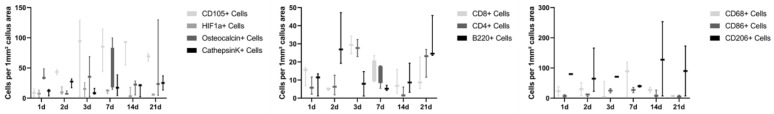
Cell counts of specific cells were performed within a certain callus area to obtain an overview of the frequency in relation to each other. For each cell type, 3 windows were analyzed, representing the callus center, margin, and adjacent bone marrow, n = 3.

## Data Availability

The datasets used and/or analyzed during the current study are available from the corresponding author upon reasonable request.

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
