# Peer review of "Complex Spatio-Temporal Interplay of Distinct Immune and Bone Cell Subsets during Bone Fracture Healing"

_cells, 2023, doi:10.3390/cells13010040_

Round 1

Reviewer 1 Report

Comments and Suggestions for Authors

The authors aimed to investigate various cell populations during the course of fracture healing. This research question is very interesting and the used staining methods are highly appropriate. The only two things I am missing are

1. a quantification of the cells over the course of the healing phases. Could the authors quantify the cell numbers per population divided by the callus area at each time point? 

2. whole callus pictures showing the distribution of the different cell populations. Could the authors add these pictures (each staining approach per each day, 1 representative mouse per time points should be sufficient)?

One minor point: could the authors add information about the used external fixation, since stiffness of the construct could also influence appearance of the measured cell populations? This should maybe also be mentioned in the discussion.

Author Response

We would like to thank the reviewer for the time to look at our manuscript. We have made the suggested changes as detailed below.

The authors aimed to investigate various cell populations during the course of fracture healing. This research question is very interesting and the used staining methods are highly appropriate. The only two things I am missing are

  1. a quantification of the cells over the course of the healing phases. Could the authors quantify the cell numbers per population divided by the callus area at each time point? 

We have added another figure depicting the quantification of the analysed cell subsets. For this we counted cells in 3 mm2 chosen from specific areas within the fracture callus, the middle of the fracture gap, the adjacent bone marrow and the an area next to the fracture gap. Analysis was done for 3 animals for each time point an cellular subset. From this data a quantitative distribution of the cell subsets can be deducted, albeit high standard deviation were seen.

  1. whole callus pictures showing the distribution of the different cell populations. Could the authors add these pictures (each staining approach per each day, 1 representative mouse per time points should be sufficient)?

As suggested we have added whole callus pictures of the stainings for one animal for each staining for each time point – in order to not overload the manuscript we choose to ad them as supplementary figures.

One minor point: could the authors add information about the used external fixation, since stiffness of the construct could also influence appearance of the measured cell populations? This should maybe also be mentioned in the discussion.

Thank you for this suggestion, we have now added a specification of the fixation stiffness in the methods and discussion part of the manuscript.

Reviewer 2 Report

Comments and Suggestions for Authors

Comments of cells-2739737

The main weaknesses of the manuscript:

1. The abstract can be improved to show more significance and findings of study.

2. The structure of the paper is not adequate. If you decide to use a results section and a discussion section, please don’t put any discussion in the results and do not put any new results in the discussion. Moreover, some results are put in an order that I cannot understand in terms of logic.

3. Scales should be established in all figures.

4. The discussion points for Figs. 4-9 are not well supported with other literature findings.

5. Please add references from the last five years.

6. What is the mechanism of bone healing?

Author Response

We would like to thank the reviewer for taking time to review our manuscript, we have addressed all the points made by the reviewer in detail.

  1. The abstract can be improved to show more significance and findings of study.

As suggested by the reviewer the abstract has been rewritten to better display the findings of the study.

  1. The structure of the paper is not adequate. If you decide to use a results section and a discussion section, please don’t put any discussion in the results and do not put any new results in the discussion. Moreover, some results are put in an order that I cannot understand in terms of logic.

Thank you for pointing out that discussion should be avoided within the result section – we have removed these were we found them. As for new results mentioned within the discussion we are not sure to which paragraph this comment referred and thus were unable to correct this mistake. The results should follow the consecutive events of the bone healing process – beginning with the pro-inflammatory reaction that immediately follows the injury, followed by the revascularization step and so on. We changed the sequel now to better adhere to this process. However, in order to not constantly switch between the multitude of analysed cell types we then analysed the occurrence of the specific cell type over the complete healing period.

  1. Scales should be established in all figures.

We apologize for this overview and have added the scale bars were necessary.

  1. The discussion points for Figs. 4-9 are not well supported with other literature findings.

We would like to thank the reviewer for pointing out the missing referneces and have added those to the manuscript

  1. Please add references from the last five years.

We have added new references throughout the manuscript were fitting.

  1. What is the mechanism of bone healing?

We tried to include a description of the bone healing process in the discussion (second paragraph). Within the manuscript, we also stated that the healing in the chosen mouse osteotomy model proceeds through endochondral ossification. The healing process has been depicted in Figure 2 and within the results section covering this figure each healing phase has been described referring to the tissues visible in the staining.

In more depth, the healing process has been described within the discussion as starting of with an inflammatory process followed by a revascularization – two very important mechanisms within the bone healing process. And throughout the discussion section the subsequent phases of the healing process are discussed, highlighting those mechanisms as pertaining towards the specific cell subsets analysed within this work.

We do hope that we have thus given an insight into the mechanism of bone healing; at least in part, as it is vastly complex, as far as the analysed cell subsets are concerned.

Round 2

Reviewer 2 Report

Comments and Suggestions for Authors

Comments of cells-2739737

The manuscript is well revised and can be published in present form.